# Recommendations of the Netzwerk Kindersimulation for the Implementation of Simulation-Based Pediatric Team Trainings: A Delphi Process

**DOI:** 10.3390/children10061068

**Published:** 2023-06-16

**Authors:** Ruth M. Löllgen, Ellen Heimberg, Michael Wagner, Katharina Bibl, Annika Paulun, Jasmin Rupp, Christian Doerfler, Alex Staffler, Benedikt Sandmeyer, Lukas P. Mileder

**Affiliations:** 1Pediatric Emergency Department, Astrid Lindgren Children’s Hospital, Karolinska University Hospital, 17164 Solna, Sweden; 2Department of Women’s and Children’s Health, Karolinska Institutet, 17164 Solna, Sweden; 3Netzwerk Kindersimulation e.V., c/o Universitätskinderklinik, 72076 Tübingen, Germany; ellen.heimberg@med.uni-tuebingen.de (E.H.);; 4Pediatric Intensive Care Unit, Department of Pediatrics, University Hospital Tübingen, 72076 Tübingen, Germany; 5Division of Neonatology, Pediatric Intensive Care and Neuropediatrics, Department of Pediatrics, Comprehensive Center for Pediatrics, Medical University of Vienna, 1090 Vienna, Austria; 6Pediatric Intensive Care Unit, Department of General Pediatrics, University Hospital Münster, 48149 Münster, Germany; 7Pediatric Intensive Care Unit, Ostalb-Klinikum, 73430 Aalen, Germany; 8Gemeinschaftspraxis Kinder und Jugendärzte G. Fleck/C. Dörfler, 93059 Regensburg, Germany; 9Division of Neonatology, Central Teaching Hospital of Bolzano/Bozen, 39100 Bolzano, Italy; 10Institut für Notfallmedizin und Medizinmanagement INM, Klinikum der Universität München, LMU München, 80336 Munich, Germany; 11Division of Neonatology, Department of Pediatrics and Adolescent Medicine, Medical University of Graz, 8036 Graz, Austria

**Keywords:** guidelines, neonatology, pediatrics, quality criteria, simulation-based training

## Abstract

Background: Serious or life-threatening pediatric emergencies are rare. Patient outcomes largely depend on excellent teamwork and require regular simulation-based team training. Recommendations for pediatric simulation-based education are scarce. We aimed to develop evidence-based guidelines to inform simulation educators and healthcare stakeholders. Methods: A modified three-round Delphi technique was used. The first guideline draft was formed through expert discussion and based on consensus (*n* = 10 Netzwerk Kindersimulation panelists). Delphi round 1 consisted of an individual and team revision of this version by the expert panelists. Delphi round 2 comprised an in-depth review by 12 external international expert reviewers and revision by the expert panel. Delphi round 3 involved a revisit of the guidelines by the external experts. Consensus was reached after three rounds. Results: The final 23-page document was translated into English and adopted as international guidelines by the Swiss Society of Pediatrics (SGP/SSP), the German Society for Neonatology and Pediatric Intensive Care (GNPI), and the Austrian Society of Pediatrics. Conclusions: Our work constitutes comprehensive up-to-date guidelines for simulation-based team trainings and debriefings. High-quality simulation training provides standardized learning conditions for trainees. These guidelines will have a sustainable impact on standardized high-quality simulation-based education.

## 1. Introduction

Serious and life-threatening pediatric emergencies occur much less frequently in clinical routine than adult emergencies [1,2]. Healthcare practitioners who work in pediatrics and pediatric acute care are therefore much less frequently exposed to real-life acute events. However, excellent care and management of pediatric emergencies and optimal patient outcomes largely depend on the effective teamwork of well-prepared and regularly trained teams [3]. Since acute events are so rare in pediatrics, pediatric acute care, and neonatology, regular simulation-based (SB) team trainings and debriefings are required for all staff looking after children, including, e.g., pediatricians, pediatric emergency physicians, pediatric intensivists, pediatric anesthetists, pediatric surgeons, midwives, prehospital and nursing staff, allied health, and primary care and adult physicians working in mixed adult or pediatric settings. Regular SB team training and debriefing sessions are essential for safe and effective team collaboration in the clinical setting of pediatric emergencies. Simulation trainings are a prerequisite for optimal patient outcomes in the emergency setting. International pediatric resuscitation guidelines (e.g., European Resuscitation Council (ERC) guidelines) formally back this up. These guidelines explicitly recommend that regular, high-quality team training is performed in all institutions to improve team performance, patient safety, and patient outcomes [4]. However, the current SB education practiced in European countries is currently situated far from that suggested goal [5]. Furthermore, published guidelines for the implementation of pediatric SB team trainings are noticeably scarce. However, such guidelines are needed in hospitals and in the prehospital setting. They should be used by a wide range of healthcare providers (HCP) looking after children and neonates. This scarcity of guidelines, in turn, further impedes the large-scale, uniform implementation of simulation education across healthcare settings. Existing recommendations predominantly stem from adult patient team trainings (German language 1, pediatric team trainings UK 1) [6,7]. There is a knowledge gap regarding guideline development for SB education. Additionally, regional and national legal requirements are heterogenous, and mostly nonexistent regarding the implementation of such high-quality SB educational activities in the curricula of medical or nursing students, or as mandatory parts of continuous professional development of junior medical and nursing trainees, prehospital staff, and specialists. Although a certain routine exists on how SB activities (in situ in hospitals or the prehospital setting, or in simulation training centers) should be implemented and run, this is mostly based on expert experience and consensus. SB education is heterogenous regarding the design of the course, instructor skills and prerequisites, and content and quality of the debriefing. This heterogeneity of SB education may lead to suboptimal, low-quality trainings, failure to achieve the desired training outcomes, and ultimately, compromised patient safety. Varying or low-quality training of simulation educators and a lack of adherence to minimal standards of delivering SB training may cause harm to participants and have negative effects on their learning and clinical performance. 

### Aim of This Study

We sought to develop comprehensive, evidence-based guidelines for pediatric SB team trainings in collaboration with Netzwerk Kindersimulation to serve as a reliable source of information for simulation educators and healthcare stakeholders who wish to follow international guideline recommendations.

## 2. Materials and Methods

### 2.1. Setting

Virtual and in-person meetings (Germany) of simulation experts from four central European countries and regions (Austria, Germany, Switzerland, and South Tyrol) were held.

### 2.2. Delphi Process

The described process was based on evidence-based principles of the Delphi process and included all four methodological characteristics of the Delphi technique, i.e., (1) a group of experts being questioned about an issue of interest, (2) using an anonymous process to avoid social pressure and conformity, (3) comprising several rounds of iterative enquiry, and (4) informing subsequent rounds based on the results of the previous round [8]. We presented the results of the ongoing process once a year at our annual network members meeting to obtain feedback from members. Although there was no formal discussion on specific content or aspects of the quality criteria, these physical meetings must be acknowledged, rendering this a modified Delphi technique [9]. The process of planning, conducting, and reporting on our study was strictly guided by published guidelines [8].

### 2.3. Guideline Development

We used a modified Delphi technique to develop quality and standard criteria for the implementation of high-quality SB trainings and debriefings in pediatric acute care in the hospital and prehospital setting. The described process comprised three iterative Delphi rounds, one which was internal among the ten panel members and two which were external. The process started in March 2017 with ten members from the Netzwerk Kindersimulation (NKS) from Germany, Austria, Switzerland, and South Tyrol; these members are also authors of this paper. All panel members are practicing experts in the field of pediatric and neonatal SB training and debriefing. All ten panel members are also clinical specialists in pediatric intensive care, pediatric emergency medicine, neonatology, or prehospital emergency care. First, an online literature search was conducted to find existing guidelines and recommendations for SB education and debriefing. Second, the scope of the quality criteria was defined through expert group discussion. After having decided on the table of contents, regular video-conference calls involving all ten panelists were used to draft a first version of the quality criteria—this was based on expert consensus, experience, and published evidence. Then, each expert was assigned one chapter and was asked to perform a more in-depth literature review and to revise the allotted chapter based on more detailed and robust evidence. These individually amended chapters were again reviewed together by the whole expert panel, resulting in the first revision of the quality criteria (Delphi round 1). Discrepancies were generally resolved by panel discussion and, if necessary, by a further literature review. After having revised the first version of the quality criteria in July 2018, each panel member was asked to suggest three to five potential reviewers for the second Delphi round. Requirements to qualify as a potential reviewer were: (1) being an expert in pediatric and/or neonatal SB education and training with a minimum of 5 years of professional expertise; (2) being from a variety of countries of practice, professions, and specialties; and (3) not being involved in the previous steps of the Delphi process. Finally, 12 experts from the United States of America (*n* = 1), Germany (*n* = 4), Austria (*n* = 2), and Switzerland (*n* = 5) were invited in September 2018 by email to engage as reviewers. Of these, all 12 agreed to serve as reviewers and signed a declaration of confidentiality. Aiming for a transparent and open-response process during the second Delphi round [10], we asked the reviewers (i) to assess the content, comprehensibility, language and grammar, redundancy of information, and structure of the first draft of the quality criteria. We then asked the reviewers (ii) to offer written feedback and/or specific qualitative suggestions for improvement, as appropriate. Written reminders were sent out by email before the suggested deadline for submission of the feedback to ensure the reception of timely responses. Once all responses were received, each individual reviewer comment was read and discussed by the expert panel and consensus was sought to accept and implement or reject the suggestion or comment. If most panelists approved the reviewer suggestion, it was implemented in the document. (Delphi round 2). For the third, more specific Delphi round, the 12 expert reviewers were sent the revised version of the quality criteria, including detailed explanations for the rejection of individual suggestions if applicable [10]. The 12 external reviewers were asked to agree or disagree with the expert panel’s decision in their own comments. If individual reviewers did not initially agree with the decision, the topic of interest was discussed within the panel between the 12 external experts until consensus was reached. In addition, all reviewers were then asked to assess the entire revised document again, now with a more targeted focus on the content and cited evidence of the document. After that final appraisal, the Delphi process was completed. Thus, consensus was reached after three rounds, with decisions made for all reviewer comments and the final version of the document by the joint group of NKS panelists and expert reviewers. Finally, the German document was formally language copy-edited by a German language teacher [11] (see Appendix A). This language-edited German version was then professionally translated into English [12] (see Appendix A).

## 3. Results

Among the twelve external reviewers, there were three anesthesiologists, two pediatric emergency medicine specialists, two neonatologists, two nursing professionals, one psychologist, one paramedic, and one airplane pilot and crisis resource management (CRM) trainer. CRM comprises a set of specific non-technical or non-medical cognitive and interpersonal skills which form the basis for and greatly enhance effective team performance [13]. The response rate for the second (external) Delphi round was 100%. Most of the 401 comments were related to language and grammar (*n* = 167, 41.6%) and content (*n* = 156, 38.9%). Categorized reviewer responses of the second Delphi round are summarized in Table 1. During the third Delphi round, four of the twelve external reviewers (33.3%) offered further suggestions for improvement, while the other eight reviewers (66.7%) approved the revised version of the quality criteria. Of the total 30 comments and suggestions, 19 (63.3%) referred to contextual aspects and 5 (16.7%) to language and grammar (Table 1).

The resulting final 23-page document begins with a preamble, outlining that the recommendations “are intended to serve as a framework for the organization, implementation and quality assurance” of simulation-based team trainings of neonatal and pediatric emergencies. The preamble is followed by eight chapters, covering the topics of: (i) general learning objectives; (ii) required skills and qualifications of simulation trainers; (iii) conditions for creating general effective learning environments; (iv) conditions for creating particular simulation environments; (v) human resources to deliver simulation-based team trainings; (vi) development and scripting of simulation scenarios; (vii) actual delivery of simulation-based trainings and debriefings; and finally, (viii) feedback and training evaluations as means of continuous program and trainer development. While the contents of the recommendations are based on expert consensus and experience, they include a total of 64 references to scientific articles from the medical education and simulation literature to back up the expert consensus, as well as to emphasize the knowledge gaps and topics for further research.

## 4. Discussion

Our work constitutes comprehensive, up-to-date, consensus- and evidence-based guidelines for the implementation of SB team trainings and debriefings. These elaborated quality standards for the implementation of SB education were translated from German [11] to English [12] for broader use across the simulation educator community. Our recommendations were adopted as international guidelines by the Swiss Society of Pediatrics (SGP/SSP), the German Society for Neonatology and Pediatric Intensive Care (GNPI), and the Austrian Society of Pediatrics (ÖGKJ). We suggest that these minimal standards for SB team trainings should be met by all institutions, simulation educators, and simulation facilitators who plan and conduct SB education. We believe that adhering to these recommendations will lead to a sustainable, psychologically safe, and non-judgmental learning experience for the individual and the team, and ultimately result in improved patient safety and patient outcomes. 

### 4.1. Required Skills and Qualifications of Simulation Trainers, Creating Effective Learning Environments, Development and Scripting of Simulation Scenarios, Actual Delivery of Simulation-Based Trainings, and Debriefing

Scenario development, debriefing of the actual simulated cases, and establishing and maintaining psychological safety in the simulation are the mainstays of any simulation process. Psychological safety is defined as a shared belief within a team that every individual member can take risks and share their opinions, beliefs, ideas, or concerns; speak up with questions; and make mistakes without fear of negative consequences for that behavior [14]. We suggest that at least one of the facilitators should be a formally trained simulation educator, so that a minimal standard of provided simulation scenarios and debriefing can be assured. There is a range of offered simulation trainer courses on the market across the globe, and the quality and standards of simulation instructor courses vary widely. To date, no formal minimal quality standard requirements have been formulated or published for simulation trainer courses. We therefore suggest that trainer candidates seek expert information with regard to which course is the best to attend. It would be desirable in the future that minimal standard criteria for such simulation trainer courses are published to inform simulation instructor candidates. The sustainable effect of SB team training depends largely on the simulation trainers’ education, experience, and their commitment to their own ongoing professional development (e.g., by attending conferences or receiving peer feedback or a debriefing of their own debriefing). Thus, their high-quality debriefing skills and a high standard of the overall SB educational activity can be maintained. Ensuring that psychological safety is created and maintained throughout the simulation and the debriefing is an essential part of the SB activity [15]. By adhering to a range of implicit and explicit strategies to maintain psychological safety, the expected learning outcomes will be achieved in a more substantial way, and no harm will come to the participants. Untrained or poorly trained simulation educators could, for example, inadvertently ask unappreciative, accusatory questions instead of using a non-judgmental debriefing technique [16]. Moreover, participants could be given closed, yes-or-no questions or “hint and hope” questions (questions formulated in a way that force participants to reply exactly with what they believe the educator has in mind). Thus, low-quality debriefings may yield minimal team reflection, leading to self-doubt and poor learning outcomes. The art of developing and scripting scenarios is taught extensively in high-quality simulation educator courses. Simulation trainers should be familiar with a range of different scenario-directing tools to be applied during the scenario. To achieve a certain learning goal (e.g., participants will be able to apply the traumatic cardiac arrest algorithm), the simulation educator in the role of the scenario director may need to help the participants on their way to their learning goal. A range of different tricks have been suggested that may be used for various groups of participants, e.g., signals, lifesavers, or noises. Signals or life savers are best described as an important piece of information delivered by a confederate (a person privy to the scenario and involved in the scenario as an actor, e.g., parent or nurse) or the scenario director. Such a hint, tip, or buzz word can be communicated to the team during the scenario if participants get stuck or steer down the wrong path in the scenario. Signals may be needed to support learners in achieving the desired patient outcome or to get the team back on track if it is on an erroneous path. An inexperienced group of participants, for example, may realize that the patient is not improving and may need a strong hint. The confederate consultant surgeon, e.g., might need to say, “Team, this is traumatic cardiac arrest, we are in the wrong algorithm!” Thus, the team can realize why the patient is not improving. In contrast, more experienced learners may not need a verbal signal, but rather a mere change in vital parameters (e.g., severe bradycardia) may be the subtle signal they need to realize they are on the wrong path. Noise is best described as a distractor that may challenge a team of more advanced participants on their way to a specific learning goal. A noise could be a low normal heart rate at 58 bpm in an unconscious infant. Advanced participants, e.g., will be challenged as to whether to commence cardiopulmonary resuscitation at this heart rate (if the child is unresponsive) or not (the child is bradycardic but moaning). These effective tools allow the simulation educators to tailor the simulation scenario to their specific target group to achieve an optimal learning outcome. The actual debriefing of a simulated scenario is an art and requires high competence. Different debriefing techniques exist, e.g., non-judgmental debriefing, where precise feedback is combined with genuine curiosity [16] or the Steinwachs model [17]. These debriefing techniques are extensively taught in trainer courses. We recommend that participants use and constantly improve their newly acquired debriefing skills in their own institutional SB team trainings or in the clinical setting and gather at least one year of experience prior to attending an advanced simulation instructor course. It is suggested that more junior simulation trainers seek regular feedback for their debriefings (debriefing of the debriefing) to constantly apply suggested improvement strategies in their coming debriefings. However, these recommendations are consensus- and experience-based and studies are needed to back them up.

### 4.2. Human Resources to Deliver Simulation-Based Team Trainings

Cost, resources, and time are major obstacles to implementing simulation education programs into student and trainee curriculums and into continuous professional development. In one survey study, however, the effectiveness of the debriefing was priced higher than cost-savings when funding was available, and instructor-led debriefing was preferred over self-debriefing [18]. These findings underline what was previously discussed. Effective debriefings, which, in turn, depend on the availability of trained simulation educators as opposed to peer debriefing or self-reflection only, and team reflection will lead to improved clinical performance immediately and in the long term. Improved clinical team behavior, in turn, will lead to less adverse events and better patient outcomes, and ultimately will lower costs for the institution and healthcare system in general. These facts need to be understood when approaching superiors and stakeholders and pitching the implementation of SB team training in individual institutions. These facts provide valuable arguments in favor of sparing no monetary, resource, or timely expenses when planning the implementation of SB educational activities. A well-formulated list of researched arguments is currently being prepared for publication by the authors of this study. This list of arguments is intended for use by healthcare practitioners when pitching the implementation of SB education in their institutions. The same arguments hold true for the costs, resources, and time required for the conduction of high-quality SB education. However, even once financial and personal resources have been provided, it will always remain a challenge to plan and fit regular SB education into the busy and often unpredictable schedule of acute care settings. The conduction of SB training should, however, always be a priority and priced as equally high as SB patient safety interest groups and meetings. The team learning and transformation of teams during and after SB educational activities is worth the financial, resource, and time-consuming efforts of implementing SB trainings and debriefings. We highlight that this described effect on teams concerns each team member, and all staff will benefit to some extent from the SB learning experience, ranging from the new and inexperienced medical student to the long-standing, eminent head of the department. 

### 4.3. Feedback and Training Evaluations as Means of Continuous Program and Trainer Development

We suggest that regular debriefings of the simulation event in general, whether in situ, in-house, or simulation-center-based, and debriefings of the debriefers are performed. This practice aims at constantly improving the skills of the debriefers and the standards of the facilitated education sessions. Specific tools for the debriefing of the debriefing are available, e.g., the Debriefing Assessment for Simulation in Healthcare (DASH) or Objective Structured Assessment of Debriefing (OSAD). Furthermore, to promote these peer debriefings beyond one’s own institution or own faculty, we have implemented a peer feedback program available to applicants on our homepage [19]. 

### 4.4. Netzwerk Kindersimulation

NKS is an international association of individual members and simulation centers in central Europe which uses German as its official language. We aim to save children’s lives through active engagement in pediatric simulation. The networking of members is our priority. We aim to promote networking and member exchange by providing easier access to human and material resources for simulations. Our target is to increase the quality and professionalism of pediatric simulations by supporting the professional and creative implementation of individual and innovative ideas to promote high-quality simulation training. We also support research projects on simulation, both content-wise and financially. We seek to be the expert point of contact for pediatric simulation training delivery through the development of evidence-based standards for simulation-based training and education in cooperation with simulation associations and interest groups in the international field. We focus on anchoring regular team training where pediatric emergencies are simulated and to demand sufficient education and training resources for pediatric simulation training. The implementation of NKS goals takes place directly through our working groups, such as, e.g., this described Delphi process. 

### 4.5. Standards and Applications of SB Trainings and CRM versus Resuscitation Algorithm Courses

Well-established standardized algorithm courses, e.g., Basic Life Support (BLS), Advanced Life Support (ALS), Neonatal Life Support (NLS), Pediatric Advanced Life Support (PALS) or European Pediatric Advanced Life Support (EPALS), are based on international guidelines, e.g., from the American Heart Association (AHA) or European Resuscitation Council (ERC), and are used to train participants in medical skills and the knowledge required for the management of critically ill neonates, children, and adolescents. In this consensus- and evidence-based reference work, however, we explicitly describe the minimal standards for SB team trainings where not only medical, but above all, non-technical skills (NTSs) are trained and debriefed in depth. NTSs include all aspects of CRM [20], e.g., situational awareness, team leader- and followership, and excellent communication. NTSs are paramount for outstanding team performance and positive patient outcomes. In one study, closed-loop communication markedly improved time-to-task completion in pediatric trauma resuscitation [21]. Another study found a significant and consistent correlation between applied teamwork behavior and compliance with neonatal resuscitation program guidelines and the quality of care in the delivery room [22]. The recommendations detailed in our guidelines do not only inform the more traditional and often more comprehensive simulation-center-based education sessions, but they are also applicable to off-site simulations using in-house training (training in hospital rooms set up for simulation separated from the clinical setting and often with less simulation equipment than facilities in a simulation center) or in situ simulation conducted in the actual clinical setting. It has been suggested that the choice of setting does not impact individual or team learning, but that in-house or in situ SB education may further organizational learning more than simulation-center-based training [23]. Certainly, regular high-quality SB team training is recommended by international resuscitation guidelines (e.g., ERC guidelines [4], and there is broad evidence to back this up [24,25,26,27,28]. Additionally, simulations have been implemented across most countries and most disciplines to date. However, this is in contrast with the scarcity of published guidelines detailing the very standards and quality criteria these trainings should meet, how to design team-based simulation training effectively, and what requirements simulation educators should meet. Only a few previously published pediatric guidelines exist [6], and most are not explicitly dedicated to the field of pediatrics [7,29] or are designed for other specific fields, e.g., nursing [30] or surgery [31]. The NKS quality criteria guidelines resulting from the Delphi process fill this knowledge gap [11,12].

### 4.6. Limitations

First, we acknowledge that this document is a living work. The continuous appearance of new evidence and new knowledge regarding SB education will require that this document needs to be brought up to date regularly. Certain aspects may require more detailed discussion in the future, e.g., the emerging field of psychological safety in simulation and distance simulation [15,32], or simulation used in competition and assessment contexts [33]. Second, this first edition of the guidelines refers to traditional simulation-based team training and debriefing only. It explicitly does not cover tele-simulation, e-simulation, remote/distance simulation, and avatar or virtual simulation. Although many recommendations may apply overlappingly to all these fields, the existing evidence for traditional SB education may not be applicable to other formats. Distance simulation is a relatively new and important emerging field which has gained new importance during and since the coronavirus (COVID-19) pandemic [33]. Last, this quality criteria pilot version has not yet been validated in the healthcare simulation context. Future validation studies are therefore required. 

## 5. Conclusions

This expert consensus reference work will greatly support simulation educators and stakeholders in the implementation of SB education. The dissemination of our comprehensive guidelines for the implementation of SB team training will not only promote the facilitation and standardization of high-quality SB education across pediatric and neonatal disciplines in hospitals and simulation centers where SB education is already taking place, but it will also enable the start up of SB educational activities where costs, resources, and time have not yet been employed and will serve as a reference work for stakeholders who may be unfamiliar with the concept of SB training. Its distribution may also be adapted for adult emergency SB training. This document is a living work that needs to be constantly updated to include newly emerging evidence. Future reference works should include recommendations for the implementation of distance simulations and the psychological safety associated with this. 

## Figures and Tables

**Table 1 children-10-01068-t001:** Categories of reviewer comments evaluated during both external Delphi rounds. Twelve external reviewers commented on items which, in their opinion, required revision, addition, or removal of content in the individual domains.

	Content of Document	Comprehensibility of Content	Language and Grammar	Redundancy of Content	Structure of Document	Total
2nd Delphi round						
Reviewer comments (*n*)	156	45	167	12	21	401
3rd Delphi round						
Reviewer comments (*n*)	19	3	5	1	2	30

## Data Availability

Data supporting the reported results can be requested from RML and LPM.

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
