# Peer review of "Recommendations of the Netzwerk Kindersimulation for the Implementation of Simulation-Based Pediatric Team Trainings: A Delphi Process"

_children, 2023, doi:10.3390/children10061068_

Round 1

Reviewer 1 Report

 The Article entitled “Recommendations of the Netzwerk Kindersimulation for the Implementation of pediatric simulation-based team training: A Delphi Process” is well-written and structured. However, a minor shortening of the “Discussion,” without missing out on any important points, would probably make it easier readable.

There are a few comments/corrections that I would like to make:

1.      The zip codes of the Affiliation addresses are missing (see Author Instructions)

2.      References should not be used in the abstract. All the numbers should be omitted from the Abstract, and all the proper corrections should be made in the correct sequence of the numbers in the text.

3.       Line 34: consisted of (and not in)

4.      Line 39-40 and lines 171-172: Use Capitals for the first letter in each word (e.g., Austrian Society of Pediatrics)

5.      Line 55-56: Rephrase “Such training is a prerequisite…”.

6.      Line 76: “harm done participants”. Either clarify/rephrase of completely omit.

7.      Line 129-130: rephrase as such: “including detailed explanations for rejection…”

8.      Line 130-131: The sentence “For each reviewer suggestion….explained” can be omitted since the context is the same as the previous sentence.

9.      Line 141-143: The percentages probably don’t add any benefit.

10.  Line 144: Please define “CRM”

11.  Line 152: “12” in the caption of Table 1 should be substituted with “twelve”

12.  Line 159-160: “…(iii)conditions for) creating effective learning environments in general and (iv) conditions for creating effective simulation environments in particular…”

13.  Line 180: “Psychological safety” should be clarified.

14.  Line 194-195: The sentence “ How to create in the literature” makes no sense

15.  Line 196-197: Please define what harm can be done to a participant during an SB training program.

16.  Line 198-202: The whole sentence needs to be rephrased, and clarifications should be made as far as “signals,” “noise”

17.  Reference section: Please correct all references using the ACS style

Minor editing required

Reviewer 2 Report

Simulation has become an integral part of medical education in Intensive Care for all age groups, but it is crucial for Pediatrics and Neonatology, where healthcare providers cannot afford to make mistakes on their patients. Being a relatively new branch of training, simulation needs appropriate guidelines, therefore the initiative of the authors to create and implement such guidelines is salutary. This being said, the purpose and relevance of the manuscript I’m being asked to review are beyond me.

Other minor points:

-          The phrase on rows 61-64 is really long and difficult to read

-          What is the meaning of “harm done participants” on row 76?

-          I’m not sure “presential” (row 81) is a real word…

-          What type of threats one would expect from simulation-based learning (row 175)?

-          Please revise the sentence on rows 194-195 – some words may have been lost there

-          Please replace “confederate”, on row 200 with a more appropriate word

-          The Discussion section overall mainly refers to simulation-based learning and not to the process of reaching a consensus through the Delphi method. The entire paragraph on rows 241-284 seems made up of separate constructs, with no meaning and no connection to each other.          

I would have greatly appreciated reading the Recommendations document in English. Unfortunately, this document was unavailable to me during peer-review, both on the link provided in the References section, as well as on engine search.

Mentioned above

Reviewer 3 Report

I read your paper and I can say you did a good job.

The paper is well-written and it revolves around the main theme described in the title and introduction. Indeed simulation-based learning in medicine is one of the safest and most efficient methods of learning.

When it comes to resuscitation, time is of the essence, and repeating the resuscitation maneuvers on a simulation-based model it s very beneficial for learning. It was proved that the medical teams that practice often resuscitation on a simulation-based model are more efficient than those who don't routinely exercise resuscitation.

So, in my opinion, your paper highlights the role of simulation in learning.

Freat job!

The English is fine

Author Response

Thank you very much for your comments. We have addressed the comments of all 3 reviewers and changed the manuscript accordingly. The changes are highlighted in the revised manuscript as per instruction.

Round 2

Reviewer 2 Report

With a few more explanations and the original Recommendations document, the manuscript makes much more sense. My initial misunderstanding came from the title – I was convinced that the manuscript should be more about applying the Delphi method to the original document.

I appreciate linking the document at the end of the manuscript, but under these circumstances, are references 11, 12 and 14, 15, 16 really necessary?

Other minor issues for me, that have more to do with editing, than anything else:

-          I wonder if row 275 is more of a subtitle and should be written in Italics.

- On rows 344-345, there is a repetition of ”excellent”, one of those instances must be replaced (e.g. ”outstanding”) and there are too many repetitions of the word ”and” – please rephrase.  
